# Ethnobotanical and Ethnopharmacological Study in the Bulgarian Rhodopes Mountains—Part _I

**Irena Mincheva [1,\*], Zheko Naychov [2], Christo Radev [3], Ina Aneva [1], Luca Rastrelli [4] and Ekaterina Kozuharova [5,\*]**

1   Institute of Biodiversity and Ecosystem Research, Bulgarian Academy of Sciences, 1000 Sofia, Bulgaria
2   Department of Surgery, University Hospital Lozenets, Sofia University St. Kliment Ohridski, 1407 Sofia, Bulgaria
3   Freelancer in Computer Sciences and Information Technologies, 4023 Plovdiv, Bulgaria
4   Department of Pharmacy, University of Salerno, 84084 Fisciano, Italy
5   Department of Pharmacognosy, Faculty of Pharmacy, Medical University of Sofia, 1000 Sofia, Bulgaria
\*   Correspondence: irena_mincheva@yahoo.co.uk (I.M.); ina_kozuharova@yahoo.co.uk (E.K.)

**Abstract:** Traditional knowledge of medicinal plants and their uses has been well documented in Bulgaria in the past. However, we know little about the contemporary traditional application of medicinal plants. Rhodopes Mountain is an ethnobotanically poorly studied region. This region is suitable for conducting field research in ethnobotany for several reasons: (i) our preliminary observation in a number of settlements revealed that the local population, in particular, relies solely on previously collected medicinal plants in winter months even in modern times; (ii) due to the relative isolation of the area, considerable authenticity of traditional methods of medicinal plant use is retained there. The aim of this study is ethnopharmacological and ethnobotanical research among the population of the Rhodopes to evaluate the contemporary use of medicinal plants. The field ethnobotanical data were collected through the in-depth method in combination with a semi-structured face-to-face interviews, adapted with modifications to the objectives of this study. The quantitative ethnobotanical index Use Value (UV) was calculated. Here we suggested a new approach in ethnobotanical research. We used nomograms to present a large volume of medicinal plants' application data, classified by the degree of their UV. This allows for a much broader view of collected and processed data. Data analyses from our filed research showed that the 92 informants mentioned utilization of a total of 114 plant species belonging to 52 families and 110 genera. The most common plants were from the families Asteraceae (16.7%), Lamiaceae (12.3%), Rosaceae (9.6%) and Amrillydaceae (3.5%), followed by Crassulaceae, Plantaginacea, Oleaceae and Solanaceae. The data presented in six nomograms revealed the most popular plants, the way of application and the corresponding medical indications in the Central and East Rhodopes, and the differences between the two sub-regions. *Sempervivum tectorum*, *Tussilago farfara* and *Plantago major* are the most often reported plants in the Central Rhodopes while these are *Cotinus coggygria*, *Prunus spinosa* and *Teucrium polium* in the East Rhodopes. The results of the study show that in the Rhodopes, the contemporary application of traditional medicinal plants is pretty much vivid. The locals in the Rhodopes still use the traditional knowledge and rely on plants to treat various health problems. They use common plants in a sustainable manner and are open to the cultivation of *Sideritis scardica*—a species which became rare after overexploitation.

**Keywords:** nomograms; traditional knowledge of medicinal plants; ethnobotany

## 1. Introduction

More than 700 species (about 20% of Bulgarian flora) are recognized as medicinal plants in Bulgaria. Most of them are native [1,2]. Some of them are rare plants [3]. Bulgaria is the second largest exporter of wild herbs in Europe and in general among the first

exporters of herbs in the world [4]. Over 80% of Bulgarian medicinal plants go for the foreign market. The great biodiversity of medicinal plants in Bulgaria creates conditions for sustainable use of these resources [5]. About 30–40 species are cultivated, but some of them are still collected from the wild [6]. Thus, many species are threatened both by excessive collection and by factors unrelated to their direct use (such as habitat loss).

Traditional knowledge of medicinal plants and their properties have been well documented in Bulgaria. The pioneer ethnobotanical data collection during the early 20th century [7,8] was followed by a period of valuable scientific publications [9–13]. Lately ethnobotanical and ethnopharmacological studies in Bulgaria are based on analyses which provide information for the medical and culinary plants popular among the population [14–17]. The results show that medicinal plants are most commonly used to treat diseases of the central nervous and musculoskeletal systems, skin, gastrointestinal and respiratory systems. Additionally, ethnobotanical data on medicinal plants that have not been documented in the literature are still being recorded [18,19]. In the last few years, there has been a trend towards the application of statistical methods that mainly examine the links between informant demographic characteristics and the use of medicinal plants [20–23]. Ethnobotanical studies aim to examine the attitudes of the population towards the use of medicinal plants, as well as to the use of products derived from them [22,23]. In these studies, awareness of medicinal plants was monitored, as well as their potential adverse effects [21] and sources of information about them [22,23]. At the same time, the proportion of studies that seek to validate available or newly documented ethnobotanical information remains low [24].

Lately, the quantitative indexes in the ethnobotanical research grow popularity as they allow easy comparison of the results obtained [25–29].

Rhodope Mountain is an ethnobotanically a poorly studied territory [30–33]. Bertsch conducted an ethno-botanical study in the Rhodopes Mountain, which aimed to gather and evaluate information on the cultural and economic importance of non-timber forest resources (medicinal plants, mushrooms) and the dynamics of their consumption in the context of the economic changes in Bulgaria [30]. The methods included surveys and semi-structured interviews with residents of the Municipality of Garmen (Western Rhodopes). This research [30] provides extensive information on the history, geography and demographic composition of both the study area and the country in the course of economic changes.

Rhodope Mountain is suitable for ethnobotanical field research because our preliminary observation in a number of settlements revealed that even nowadays, and particularly in the winter months, the local population relies solely on the medicinal plants collected during the summer. Due to the relative isolation of the area, considerable authenticity is retained about the traditional methods of medicinal plant use. Phyto-climatological and ecological differences exist in the study area with a relative geographical unity and isolation [34]. Additionally, according to the sporadic epidemiological research in the Rhodope Mountain, the mountain is a location inhabited by long living people [35]. At the same time, a study has shown that the arterial hypertension frequency among the advanced and old age population in four villages in the Rhodope Mountains is high [36].

Recently, we have been witnessing an increasing and alarming rate of biodiversity loss. According to the Food and Agriculture Organization of the United Nations, 60% of the world's ecosystems are disturbed or used in an unsustainable manner. In the European Union, only 17% of habitats and species and 11% of key ecosystems protected by EU legislation are in satisfactory condition (EEA Technical report, 12/2010). The heterogeneous climatic and geological conditions define Bulgaria as one of the richest countries in Europe in terms of biodiversity with 4100 species of vascular plants [37]. Numerous endemic and relict species of plants grow in Bulgaria; 170 species are Bulgarian and 270—Balkan endemics [38].

This study aims to conduct ethnopharmacological and ethnobotanical research among the population of the Rhodope Mountain in order to evaluate the contemporary application of the traditional knowledge of medicinal plants.

## 2. Materials and Methods

### 2.1. Study Sites, Data Collection and Pre-Processing

Geographical maps used in this research were produced thanks to the following projects: OpenStreetMap [39] (layered maps with data for the municipalities, administrative districts and state borders), BGtopoVJ (Bulgarian topographical maps and data) [40] and BGMountains (more precise Bulgarian mountains relief topographical data) [41]. The settlement names and the number of interviewed informants are shown on such a map (Figure 1).

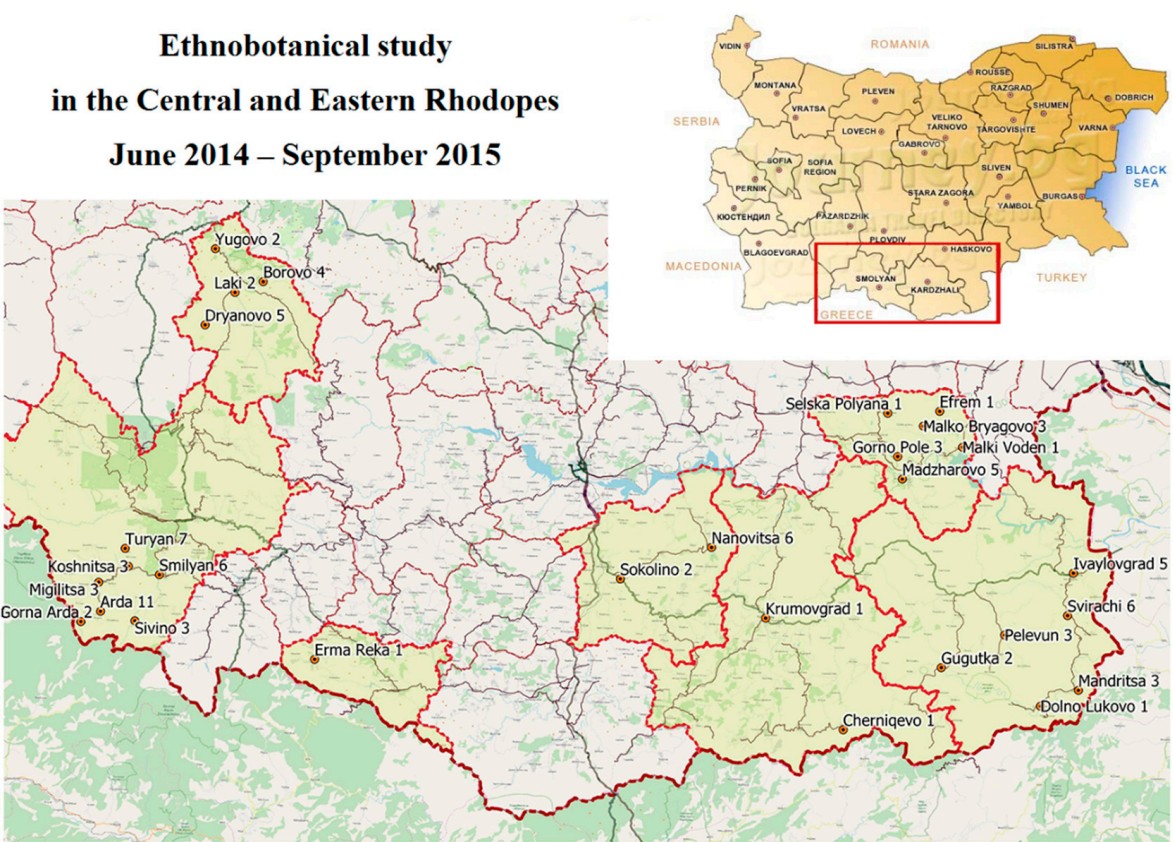

**Figure 1.** Settlements of the survey in the Rhodope Mountain and the number of informants.

A field ethnobotanical survey was conducted among the population of 29 settlements, in seven municipalities in the Rhodopes, located on the territory of four administrative districts (Plovdiv, Smolyan, Kardzhali and Haskovo); namely, 12 settlements in the Central Rhodopes (11 villages and 1 city) and 13 settlements in the East Rhodopes (13 villages and 3 cities). The settlements for the study were selected to fit into the two floristic sub-regions of the Rhodopes, Central (700–1000 m) and East Rhodopes (altitude 0–500 m) [37]. Our field study covers six Natura 2000 protected areas as follows: BG0001031 in the Central Rhodopes and BG0002014, BG0002019, BG0001032, BG0002071 and BG0002092 in the East Rhodopes [42,43]. During data collection and analyses, special attention was given to the fact that many plants in the area have conservation status [38].

The ethnobotanical survey in Rhodope Mountain was conducted during June 2014–September 2015. Ethnobotanical information on medicinal plants was collected among 92 informants during June 2014–September 2015. Additionally, information about cultivation of selected species of plants was collected in May 2021.

Residents of the Rhodope Mountains, aged over 18 years, were interviewed after prior consent. The informants were selected according to the "snowball" method—the first informant in the village is randomly appointed, and the following ones are recruited on the basis of information and contacts provided by the first informant. The "snowball"

technique provides opportunity to study the lifestyles and attitudes of hard-to-reach groups of society, which usually stay aside from sociological studies [44].

The field ethnobotanical data were collected through the in-depth method in combination with a semi-structured face-to-face interview *a modo* [45], adapted with modifications to the objectives of this study. The semi-structured interview has a lower degree of structuring, which provides a higher intensity of communication with the informants. This way of information gathering allows to examine in detail the specifics and diversity of the context to which the information relates. The in-depth interview has a drilling, expert character and is conducted in preparation for a quantitative, representative study [46]. The introductory part of the interview was about gathering demographic information about the informants—age, education and occupation. The interview follows a set of questions tested in previous field studies [17]. The questionnaire pointed to broad organ- and therapy-based use-categories [47] but was based, in general, on the International Classification of Diseases of the World Health Organization (WHO, 2018) [48]. Diseases, symptoms or conditions reported by informants included the following: abscess, warts, skin inflammation, wounds, vision, ear pain, gastritis, ulcer, diarrhea, biliary inflammation, jaundice, vomiting, headache, relaxing, cough, sinusitis, low stamina, diabetes, anemia, high blood pressure, "blood purification", anticoagulant, varicose veins, cardiac diseases, hemorrhoids, cancer, breast cancer, cervical cancer, low back pain, joint pain, trauma, abortion, childlessness, "women's diseases", mastitis, potency (sexual), hormonal imbalance, renal diseases, enuresis, prostatic adenoma, cystitis, fever, cold, toothache, hair strengthening and hernia.

The questions used in the interviews were arranged as disease/condition and plant used. These are so-called "open-ended" questions, with no fixed answers. This allows the informants to prioritize the important plant species for him/her. The diseases were named in a way that is understandable by the informants. For example, "What used to treat high blood pressure?" instead of "What used to treat arterial hypertension?".

The questionnaire collected detailed data on the plant species, its local name, the part in use, method of preparation and administration (including therapeutic or prophylactic), plant substance origin (wild or cultivated), as well as personal experience with the achieved medicinal and/or undesirable effect. The anecdotal reports were evaluated for authenticity (family traditions, healers or other). Only authentic information was considered for further data processing. The interviews were recorded with audio recorder. Vouchers were collected (kept in personal collection) or photographs were used to verify the plant identity. Identification was performed after Flora of People Republic of Bulgaria [49].

The audio recordings of the anecdotal reports were decrypted and the information was recorded in spreadsheets (Microsoft Excel 2010). Each table contains information about the informant (age, gender, location, education and occupation) and the plants mentioned. Each plant species was recorded with its Latin name, local name, the plant part and the category of its use (therapeutic or prophylactic), as well as the relevant subcategories—the specific disease to which it is applied, method of preparation (infusion, decoction, compress and ground fresh plant parts) and administration (internal, external, topical, gargle and baths). Data from all informants were listed in tables for each of the regions (Central and East Rhodopes) and then summarized in Table 1 according to Euro+Med PlantDatabase (2006–2022) [50], and notes about conservation status were added. The so-called related categories show the relationship between use for a particular disease, the way it is treated, and the way each plant is administered (e.g., "cough, infusion" and "cough, compress" are two separate related categories). Here, are analyzed only the reports for human application.

*2.2. Data Processing*

The following quantitative ethnobotanical indices were used in this study: **U**se **V**alue (**UV**).

The index **UV** is a quantitative characteristic of the relative importance of the plant species that the informant community knows and uses [51]. This value is calculated using the following formula:

$$UV = \Sigma U/n$$

**U** is the number of references to a particular plant species and **n** is the total number of informants. The **UV** index is high when there are many references to the use of a plant and tends to zero when there are few references. However, **UV** does not differentiate whether the plant is used for one or more than one disease [52,53].

**Table 1.** Plants used in the Central and East Rhodopes legend: Special regime of collection (MOEW)—Ministry of Environment and Waters restricts collection quota each year, Bal. Balkan Endemic, RDB-Red Data Book of Bulgaria.

| Plant Species | Local Name | Conservation Status |
|---|---|---|
| **Amaranthaceae** | | |
| *Amaranthus retroflexus* L. | Щир | |
| *Chenopodium foliosum Asch.* | Гърличава трева, свински ягоди, | |
| **Amaryllidaceae** | | |
| *Allium cepa* L. | Лук, суганов лук | |
| *Allium porrum* L. | Праз лук | |
| *Allium sativum* L. | Чесън | |
| *Galanthus nivalis* L., *Galanthus elwesii Hook.* | Кокиче | EN- IUCN, RDB, Protected by Biodiversity Act of Bulgaria [54] |
| *Leucojum aestivum* L. | Кокиче, блатно | Spelial regim of protection by Biodiversity act of Bulgaria [54] |
| **Anacardiaceae** | | |
| *Cotinus coggygria Scop.* | Тетра, смрадлика | |
| **Apiaceae** | | |
| *Eryngium campestre* L. | Ветрогонче | |
| **Apocynaceae** | | |
| *Nerium oleander* L. | Зокум | |
| **Araceae** | | |
| *Arum maculatum* L. | Змиарник | |
| **Araliaceae** | | |
| *Hedera helix* L. | Бръшлян | |
| **Asparagaceae** | | |
| *Ruscus aculeatus* L. | Див чемшир | |
| **Asteraceae** | | |
| *Achillea millefolium* L. | Равнец бял | |
| *Agrimonia eupatoria* L. | Камшик | |
| *Arctium lappa* L. | Рьопел, репей | |
| *Artemisia absinthium* L. | Пелин бял | |
| *Artemisia vulgaris* L. | Пелин | |

**Table 1.** *Cont*.

| Plant Species | Local Name | Conservation Status |
|:---:|:---|:---:|
| *Calendula officinalis* L. | Невен | |
| *Carduus nutans* L. | Гингер | |
| *Centaurea cyanus* L. | Синчец | |
| *Cichorium intybus* L. | Синя жлъчка | |
| *Cirsium arvense* (L.) *Scop.* | Паламида | |
| *Crepis* spp. | Брадавично биле | |
| *Helianthus annuus* L. | Слънчоглед | |
| *Matricaria chamomilla* L. | Лайка | |
| *Onopordum acanthium* L. | Магарешки бодил | |
| *Tagetes erecta* L. | Турта | |
| *Taraxacum officinale F. H. Wigg.* | Глухарче | |
| *Tussilago farfara* L. | Подбел | |
| **Betulaceae** | | |
| *Corylus avellana* L. | Леска | |
| *Pulmonaria officinalis* L. | Медуница | |
| **Brassicaceae** | | |
| *Brassica nigra* (L.) *K.Koch.* | Синап | |
| *Brassica oleracea* L. | Зеле | |
| *Sinapis alba* L. | Синап | |
| **Caryophyllaceae** | | |
| *Silene vulgaris (Moench) Garcke* | Скрипалец | |
| *Stellaria media* (L.) *Cirillo* | Звездица | |
| **Cornaceae** | | |
| *Cornus mas* L. | Дрян | |
| **Crassulaceae** | | |
| *Sedum album* L. | Брадавично биле | |
| *Sedum spectabile* L. | Дебела мара | |
| *Sempervivum tectorum* L. | Бабин квас, ушно биле | |
| **Cucurbitaceae** | | |
| *Cucurbita maxima Duchesne.* | Тиква | |
| *Ecballium elaterium* (L.) *A.Rich.* | Луда краставица | |
| **Cupressaseae** | | |
| *Juniperus communis* L. | Хвойна | |
| **Equisetaceae** | | |
| *Equisetum arvense* L. | Хвощ | |
| **Ericaceae** | | |
| *Vaccinium myrtillus* L. | Боровинка черна | |
| *Vaccinium vitis-idaea* L. | Боровинка червена | |
| **Euphorbiaceae** | | |
| *Ricinus communis* L. | Кърлеж | |

**Table 1.** *Cont.*

| Plant Species | Local Name | Conservation Status |
|---|---|---|
| **Fabceceae** | | |
| *Phaseolus vulgaris* L. | Фасул | |
| **Fabaceae** | | |
| *Astragalus glycyphyllos* L. | Клин | |
| **Fagaceae** | | |
| *Quercus cerris* L. | Цер | |
| **Gentianaceae** | | |
| *Centaurium erythraea Rafn.* | Кантарион червен | |
| **Geraniaceae** | | |
| *Geranium macrorrhizum* L. | Здравец | |
| *Pelargonium zonale* (L.) *L'Hér.* | Индрише | |
| **Gesneriaceae** | | |
| *Haberlea rhodopensis Friv.* | Орфеево цвете | Bal., Protected by Biodiversity act of Bulgaria [54] |
| **Hypericaceae** | | |
| *Hypericum perforatum* L. | Кантарион жълт | |
| **Juglandaceae** | | |
| *Juglans regia* L. | Орех | |
| **Lamiaceae** | | |
| *Clinopodium vulgare* L. | Котешка стъпка | |
| *Melissa officinalis* L. | Маточина | |
| *Menta* spp. | Мента | |
| *Mentha spicata* L. | Гьозум | |
| *Micromeria dalmatica Benth.* | Бяла мента, планинска мента | Bal. |
| *Ocimum basilicum* | Босилек | |
| *Origanum vulgare* L. subsp. *Vulgare* | Риган, балкански риган | |
| *Origanum vulgare* subsp. *Hirtum (Link) Ietsw.* | Риган бял | Collection for trading forbidden |
| *Salvia verticillata* L. | Прешленеста какула | |
| *Sideritis scardica Griseb.* | Триградски чай, Мурсалски чай | Collection for trading forbidden |
| *Stachys officinalis* (L.) *Trevis.* | Ранилист | Special regime of collection (MOEW) |
| *Teucrium chamaedrys* L. | Подъбиче червено | |
| *Teucrium polium* L. | Подъбиче бяло | |
| *Thymus* spp. | Мащерика, оленица | |
| **Liliaceae** | | |
| *Lilium rhodopeum Delip.* | Крем родопски | Bal., CR-IUCN, RDB, protected by Biodiversity act of Bulgaria [54] |

**Table 1.** *Cont.*

| Plant Species | Local Name | Conservation Status |
|---|---|---|
| **Malvaceae** | | |
| *Malva sylvestris* L. | Слез, „ебе гюмеджи" | |
| *Tilia cordata Mill.* | Липа | |
| **Moraceae** | | |
| *Morus* spp. | Черница | |
| **Oleaceae** | | |
| *Fraxinus ornus* L. | Мъждрявка, мъждян | |
| *Olea europaea* L. | Маслина | |
| *Syringa vulgaris* L. | Люляк | |
| **Orchidaceae** | | |
| *Orchis* sp. div. *Anacamptis* sp. div. *Dactylorhiza* sp. div. | Салеп | Species with various IUCN status—some of them protected by the Biodiversity Act of Bulgaria [54]; collection for trading forbidden for all of them [31] |
| **Papaveraceae** | | |
| *Chelidonium majus* L. | Саралокаво биле, префръкница | |
| *Papaver rhoeas* L. | Кадънка | |
| **Pinaceae** | | |
| *Pinus* spp. | Бор | |
| **Plantaginaceae** | | |
| *Digitalis lanata Ehrh.* | Зъбаво биле | |
| *Plantago major* L. | Петрожилка широка | |
| *Plantago minor Fr.* | Петрожила тясна | |
| **Poaceae** | | |
| *Triticum vulgare Vill.* | Жито | |
| *Zea mays* L. | Царевица | |
| **Polygonaceae** | | |
| *Polygonum hydropiper* L. | Пипеиче водно | |
| **Portulacaceae** | | |
| *Portulaca oleracea* L. | Тученица | |
| **Primulaceae** | | |
| *Primula veris* L. | Иглика | Special regime of collection (MOEW) |
| **Ranunculaceae** | | |
| *Clematis vitalba* L. | Повет | |
| *Helleborus odorus Waldst. & Kit. Ex Willd.* | Кукуряк | |
| **Rhamnaceae** | | |
| *Paliurus spina-christi Mill.* | Парички, карачелия | |

**Table 1.** *Cont.*

| Plant Species | Local Name | Conservation Status |
|---|---|---|
| **Rosaceae** | | |
| *Alchemilla* spp. (*vulgaris complex*) | Цариче | Special regime of collection (MOEW) |
| *Crataegus monogyna Jacq.* | Глог | |
| *Fragaria vesca* L. | Ягода дива | |
| *Malus pumila Mill.* | Ябълка | |
| *Potentilla erecta* (L.) *Räusch.* | Троши каменче | |
| *Potentilla reptans* L. | Петопръстник, влачещите пет пръста | |
| *Prunus persica* (L.) *Batsch* | Праскова | |
| *Prunus spinosa* L. | Трънка | |
| *Rosa canina* L. | Шипка | |
| *Rosa multiflora Thunb.* | Трендафил | |
| *Rubus fruticosus* L. | Къпина | |
| *Rubus idaeus* L. | Малина | |
| **Rubiaceae** | | |
| *Galium verum* L. | Еньовче | |
| **Salicaceae** | | |
| *Salix alba* L. | Върба | |
| **Santalaceae** | | |
| *Viscum album* L. | Имел, имала, омела | |
| **Solanaceae** | | |
| *Lycopersicon esculentum Mill.* | Домат | |
| *Nicotiana tabacum* L. | Тютюн | |
| *Physalis alkekengi* L. | - | |
| *Solanum tuberosum* L. | Картоф | |
| **Tropaeolaceae** | | |
| *Tropaeolum majus* L. | Латинка | |
| **Urticaceae** | | |
| *Urtica* sp. div. | Коприва | |
| **Viburnaceae** | | |
| *Sambucus ebulus* L. | Нисък бъз, султан | |
| *Sambucus nigra* L. | Бъзлян, висок бъз | |
| **Vitaceae** | | |
| *Vitis vinifera* L. | Лоза | |
| **Zygophyllaceae** | | |
| *Tribulus terrestris* L. | Бабини зъби | |

*2.3. Data Visualization by Nomograms*

Nomograms were used to represent large volumes of data for the usage of medicinal plants classified by the degree of their usability. The horizontal axis of the nomograms

depicts the conditions for which the informants apply the medicinal plants, as well as the way of processing the plant raw material. The vertical axis shows the medicinal plants, the part used (Herba, Folium, etc.) and the origin of the species (wild or cultivated). Two major groups of nomograms were prepared for Central and East Rhodopes. Separate nomograms were drawn up for each of the cited regions for human and prophylactic applications of medicinal plants. The mathematical processing of the data as well as the preparation of the nomograms in the present study was performed using Microsoft Excel software.

The method proposed in this work for displaying data from studies of plant species and their applications in the form of a nomogram was extended by combining the main categories (plants and applications) with three additional factors. The plant part (root, stem, leaf, flower, etc.) that is used in the said application and the source of its collection (wild and/or cultivated) was added to the species of the plant. The method of preparation (cold extract, infusion, direct usage, etc.) was added to each cited application. This allows for a much broader view of collected and processed data.

The nomogram allows for quick determination of what applications a plant has and what plants the informants in the area use to treat a particular problem. The interrelationships of a plant and an application, the part of the plant used for that application and how it is used are easily detectable. Furthermore, the displayed **ΣU** values provide information on the degree of usage. It is easy to find the links with the highest **ΣU** values. The plants and the categories are sorted in alphabetical order.

The full volume of data is presented on big nomograms (Supplementary Figures S1 and S2). Filtering was applied to reduce the big number of categories on both axes and high percentage of the elements in the nomogram without references (**ΣU** = 0) and data are presented in Figures 5, 6, 8 and 9.

## 3. Results and Discussion

### 3.1. Demographic Characteristics of Informants

The ethnobotanical survey in the Rhodope Mountain was conducted among 92 informants (73 women and 19 men) at the average age of 65 years. The selection of informants on the basis of age and gender was focused on women around and over 55 years old who are considered to have traditional knowledge and play a key role in maintaining the health of family members. The male/female ratio is similar for both regions.

The percentage of informants in the age group of 60–70 years (40.2%) is the highest, followed by the 70–80 age group (22.8%). The age distribution of informants in the two regions is similar (Figure 2).

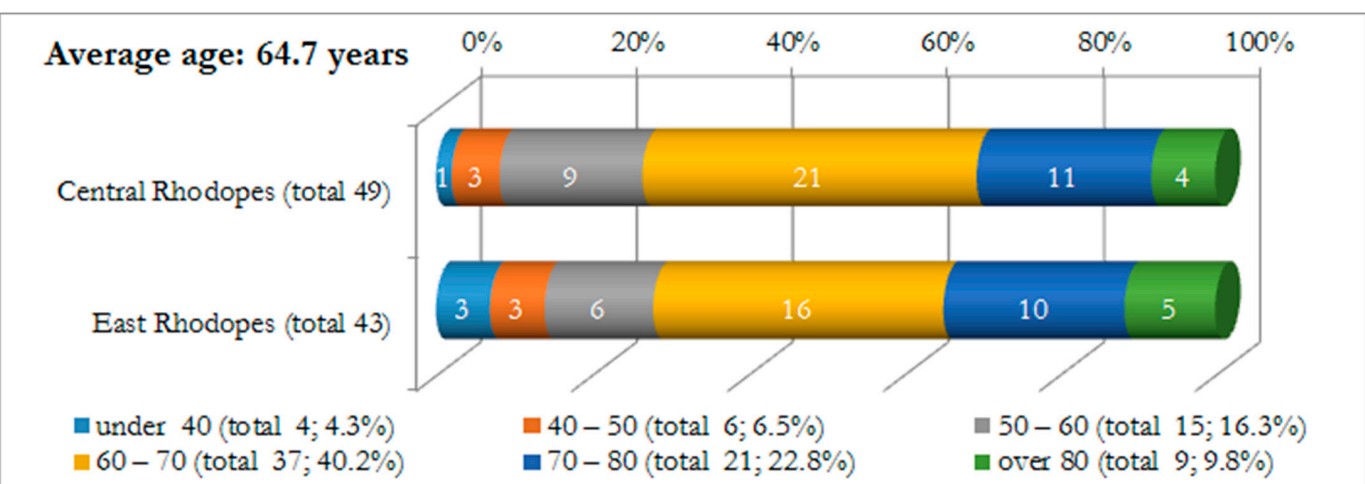

**Figure 2.** Demographic characteristics of informants by region and age.

The distribution of the informants regarding their education (Figure 3) is similar for the two study sub-regions of the Rhodopes. The percentage of informants with secondary education (46.7%) is the highest, followed by informants with primary education (33.7%).

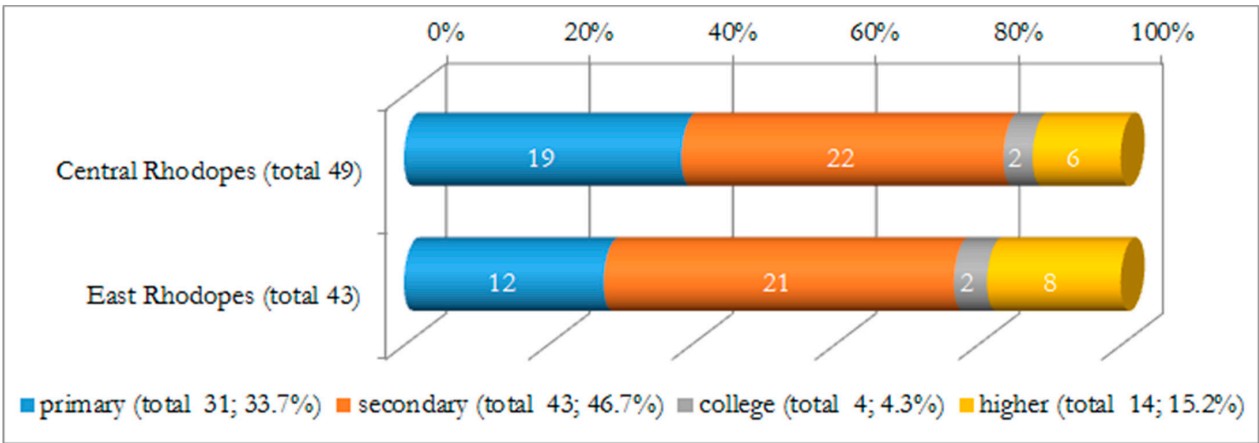

**Figure 3.** Demographic characteristics of informants by region and education.

### 3.2. Ethnobotanical Data from the Field Study

A total of 114 plant species belonging to 52 families and 110 genera were reported for human medicinal purposes (Table 1; note: the informants did not specify the orchid species, they were asked about rare species, e.g., *Lilium rhodopaeum*, but they responded that they did not use them).

Most common were species from the families Asteraceae (19 plant species or 16.7%), Lamiaceae (14 plant species or 12.3%), Rosaceae (11 plant species or 9.6%), Liliaceae (4 plant species or 3.5%), followed by Crassulaceae, Plantaginacea, Oleaceae and Solanaceae presented each by 3 plant species (or 2.6%), and the remaining 44 families include 54 species (47.4%), with two or one plant species belonging to each (Figure 4).

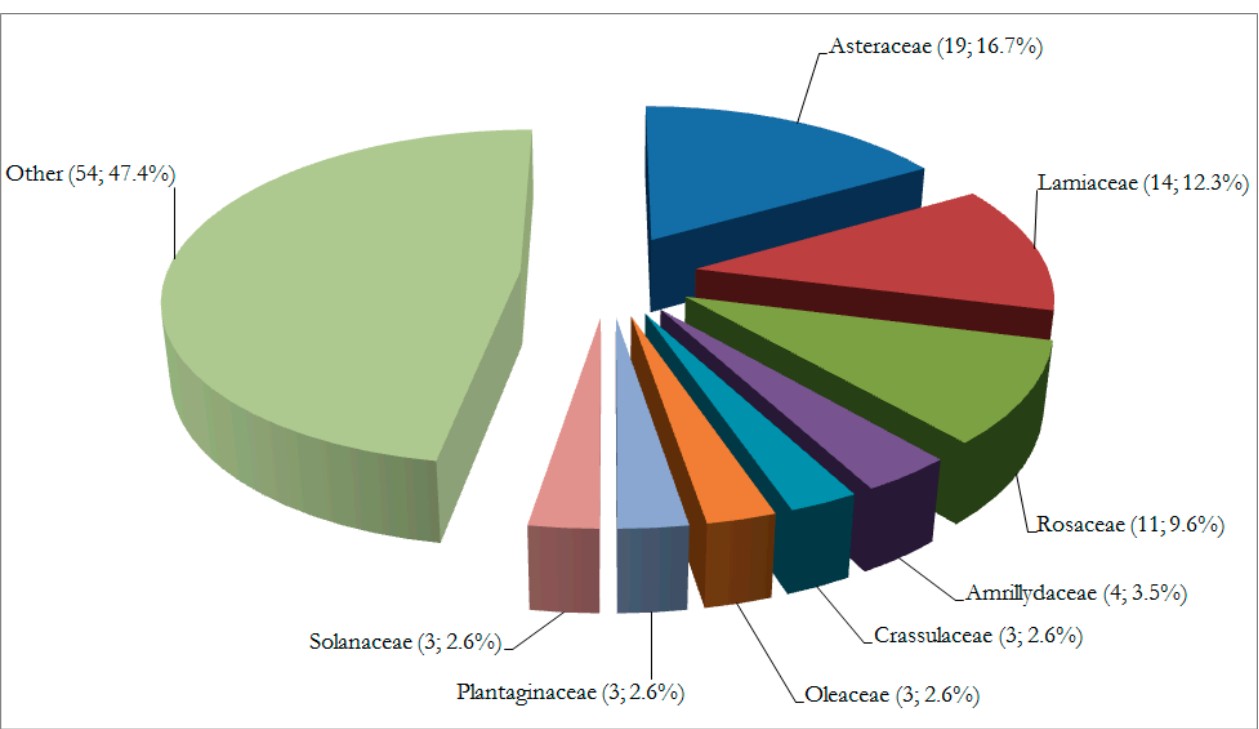

**Figure 4.** The most frequently mentioned families, depending on the number of species, Others include all families represented with less than 3 species.

### 3.2.1. The Central Rhodopes

The full nomogram of data for the application of medicinal plants classified by the degree of their usability, namely for treating human health conditions is presented in Supplementary Figure S1. Figure 5 shows only the essential part of these results after applying a filter due to the very large volume of the initial data. In this particular case, the algorithm shows only elements with **ΣU** value above 5 (or **UV** > 0.1). Thus, the obtained nomogram (Figure 5) provides the essential part of the results (205 out of 527 references).

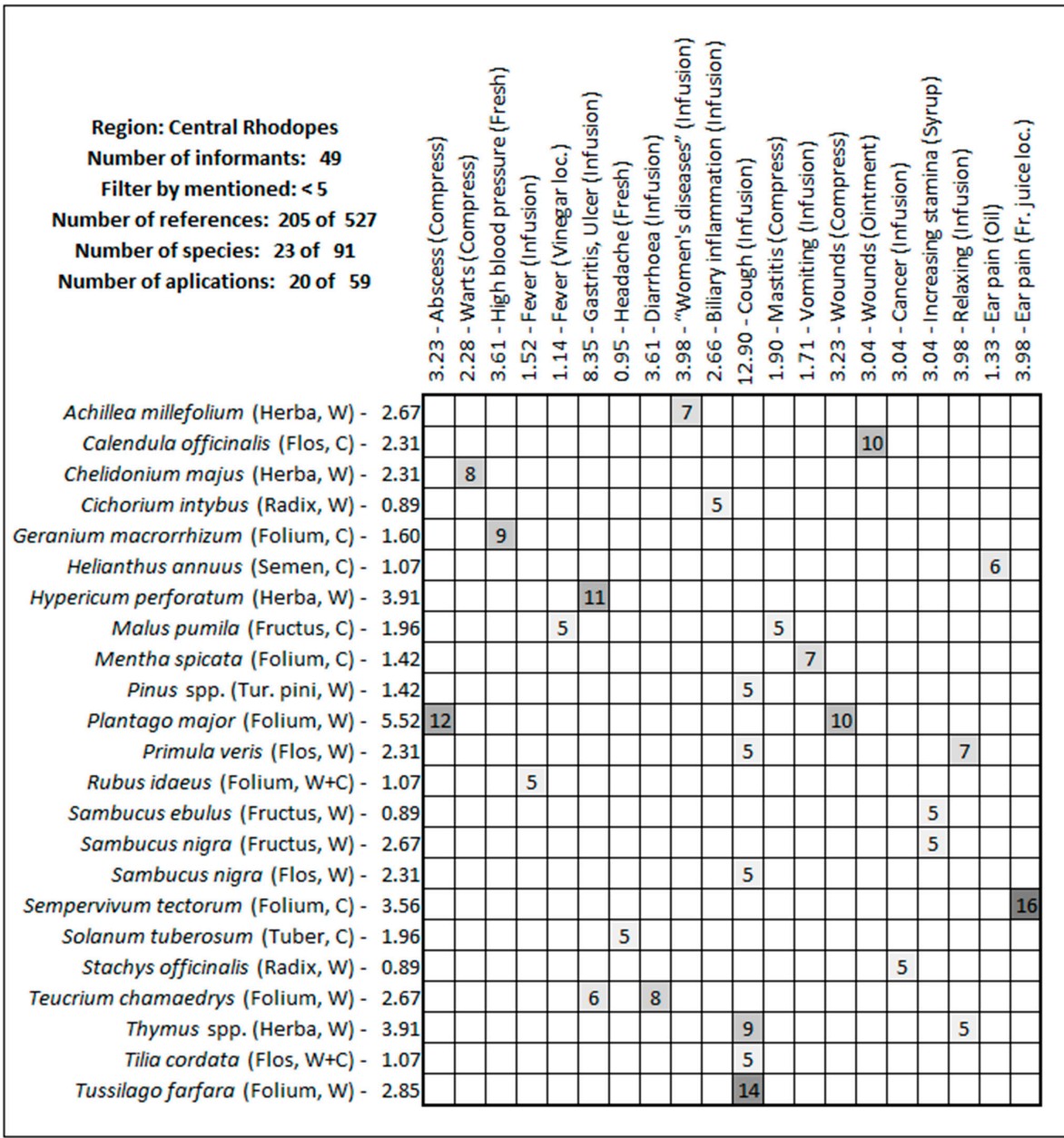

**Figure 5.** Nomogram of the medicinal plants for human use (at **ΣU** > 4) in the Central Rhodopes. Legend: abscissa/horizontal axis of the nomograms depicts the conditions for which the informants apply the medicinal plants, as well as the way of processing the plant raw material; ordinate vertical axis shows the medicinal plants *Genus species* (used part, wild/cultivated)—UV (%).

The nomogram (Figure 5), shows that the leaves of cultivated *Sempervivum tectorum* used for ear pain, prepared as fresh juice and topically applied (16), have the highest ΣU value. Also, a high **ΣU** value has the application of leave infusion of wild *Tussilago farfara*

against cough (14) and leaves of wild *Plantago major* for topical application in abscesses (12). All species listed in the nomogram are not used for more than 2 different diseases.

Other important conclusions can be drawn from the frequency of anecdotal reports. For example, leaves of wild *Plantago major* are mentioned in 5.52% of cases of abscesses cured by compress (12) and wounds (10). Cough treatment was mentioned in 12.9% of cases using infusion of wild *Pinus* spp./Pini Turiones (5), wild *Primula veris*/Primulae veris Flos (5), wild *Sambucus nigra*/Sambuci nigrae Flos (5), wild *Thymus* spp./Thimi Flos (9), wild or cultivated *Tilia cordata*/Tiliae Flos (5) and wild *Tussilago farfara*/Farfarae folium (14). The use of cough infusion is an exception because it is treated with 6 different types, while all other applications are associated with the use of no more than 2 species, which is associated with the higher significance of the disease.

Figure 6 shows the nomogram of medicinal plants with prophylactic use. It shows that informants in the Central Rhodopes use 32 plant species for this purpose. The highest relative frequency of mentions is the use of herbal tea plants (68.5%). The most mentioned are *O. vulgare* subsp. *Vulgare* (14), *Thymus* spp. (12), *Clinopodium dalmaticum* (9) and *Hypericum perforatum* (8) (Figures 6 and 7).

A popular way of using medicinal plants among the population in the Central Rhodopes is to prepare a cold extract (macerate) applied as an aromatic drink. Most commonly such cold extracts are prepared from the inflorescences of *Sambucus nigra* (6) and the inflorescence of *Melissa officinalis* (3).

In the Central Rhodopes, 80 plant species are used for treatment of various health conditions according to the informants. The degree of their use according to the value of the UV index is presented on Figure 8. This figure shows that the UV values vary from 0.02 to 0.67. The most often mentioned plants in order of degree were *Plantago major* (0.67), *Sambucus nigra* (0.60), *Hypericum perforatum* (0.46), *Thymus* spp. (0.46), *Sempervivum tectorum* (0.42), *Teucrium chamaedrys* (0.40), *Tussilago farfara* (0.33), *Achillea millefolium* (0.31), *Origanum vulgare subsp. Vulgare* (0.29). The Balkan endemic *Micromeria dalmatica* (0.17) appears to be rather popular too. Interestingly, *Sideritis scardica* has locally, traditionally low popularity and respectively low UV values (0.04), despite the fact that its popularity in Bulgaria has increased so much during the last decades that cultivation became necessary [55].

3.2.2. East Rhodopes

The full nomogram of data for the usage of medicinal plants classified by the degree of their application, namely for treating human health conditions, is presented in Supplementary Figure S2. Figure 9 shows the essential part of these results after applying a filter due to the very large volume of the initial data. In this particular case the algorithm shows only elements with $\Sigma U$ value above 3 (or **UV** > 0.1). Thus, obtained nomogram (Figure 9) provides the essential part of the results (100 out of 277 references).

The most reported is the use of a leaf of *Cotinus coggygria* for wound rinsing (8), a fruit of *Prunus spinosa* for strengthening the heart function by infusion (6) and the aerial parts of *Teucrium polium* for diarrhea (6). In 10.11% of the cases, cough treatment by infusion of *Paliurus spina-christi* fruit (4), *Tussilago farfara* leaf (4) and *Tilia cordata* inflorescence (3) were reported. The use of a leaf infusion of *Cotinus coggygria* for wound rinsing (8) and for the treatment of toothache by topical application (4) was reported with high frequency (6.5%).

According to the informants in the studied settlements in the East Rhodopes, 68 plant species are used for treatment of various health conditions. The degree of their use according to the value of the UV index is presented on Figure 11. This figure shows that the UV values vary from 0.02 to 0.42. The most often mentioned plant in order of degree were *Cotinus coggygria* (0.42), *Teucrium polium* (0.26), *Sambucus nigra* (0.23), *Chelidonium majus* (0.21), *Matricaria chamomilla* (0.21), *Paliurus spina-christi* (0.21), *Plantago major* (0.19) and *Urtica* spp. (0.19). The lower values of the popular plant species in the East Rhodopes compared to the Central Rhodopes are due to the smaller number of species mentioned in this region.

| Region: Central Rhodopes<br>Number of informants: 49<br>Number of references: 127<br>Number of species: 32<br>Number of aplications: 6 | 16.54 - Beverage (Extracted) | 55.12 - Beverage (Tea) | 3.94 - Beverage (Tea (1)) | 3.94 - Beverage (Tea (2)) | 5.51 - Beverage (Tea (3)) | 14.96 - Others |
|---|---|---|---|---|---|---|
| Achillea millefolium (Herba, W) - 0.79 | | 1 | | | | |
| Amaranthus retroflexus (Folium, W) - 1.57 | | | | | | 2 |
| Cirsium arvense (Folium, W) - 0.79 | | | | | | 1 |
| Crataegus monogyna (Fructus, W) - 0.79 | | | | | 1 | |
| Fragaria vesca (Folium, W) - 0.79 | | | 1 | | | |
| Hypericum perforatum (Herba, W) - 6.30 | | 8 | | | | |
| Malus pumila (Fructus, C) - 0.79 | | | | | | 1 |
| Melissa officinalis (Herba, W) - 3.15 | 3 | | | | 1 | |
| Micromeria dalmatica (Folium , W) - 7.09 | | 9 | | | | |
| O, vulgare subsp, hirtum (Herba, W) - 0.79 | | 1 | | | | |
| O, vulgare subsp, vulgare (Herba, W) - 13.39 | | 14 | 1 | 1 | 1 | |
| Phaseolus vulgaris (Semen, C) - 0.79 | | | | | | 1 |
| Pinus spp. (Cone, W) - 2.36 | 1 | | | | | 2 |
| Primula veris (Flos, W) - 4.72 | | 6 | | | | |
| Rosa canina (Fructus, W) - 6.30 | | 4 | 2 | | 1 | 1 |
| Rosa multiflora (Flos, C) - 0.79 | 1 | | | | | |
| Rubus fruticosus (Folium , W) - 2.36 | | | 1 | 1 | 1 | |
| Rubus fruticosus (Fructus, W) - 0.79 | | | | | | 1 |
| Rubus idaeus (Folium , W+C) - 2.36 | | 1 | | 1 | 1 | |
| Rubus idaeus (Fructus, W+C) - 0.79 | 1 | | | | | |
| Sambucus ebulus (Fructus, W) - 0.79 | 1 | | | | | |
| Sambucus nigra (Fructus, W) - 2.36 | 3 | | | | | |
| Sambucus nigra (Flos, W) - 8.66 | 6 | 4 | | 1 | | |
| Sideritis scardica (Herba , C) - 4.72 | | 5 | | | 1 | |
| Silene vulgaris (Folium , W) - 3.94 | | | | | | 5 |
| Taraxacum officinale (Flos, W) - 2.36 | 3 | | | | | |
| Thymus spp. (Herba, W) - 11.02 | 1 | 12 | | 1 | | |
| Tilia cordata (Flos, W+C) - 3.94 | | 5 | | | | |
| Triticum vulgare (Semen , C) - 0.79 | | | | | | 1 |
| Urtica spp. (Folium, W) - 1.57 | | | | | | 2 |
| Vaccinium myrtillus (Fructus, W) - 1.57 | 1 | | | | | 1 |
| Zea mays (Fructus, C) - 0.79 | | | | | | 1 |

1) winter tea: oregano, rose hips, leaves of strawberry and blackberries
2) combined tea: oregano, elderberry, raspberry and blackberry leaf
3) combined tea: oregano, lemon balm, Trigrad tea, hawthorn, rosehip, raspberry and blackberry

**Figure 6.** Nomogram of the medicinal plants with prophylactic application in the Central Rhodopes. Legend: abscissa/horizontal axis of the nomograms depicts the conditions for which the informants apply the medicinal plants, as well as the way of processing the plant raw material; ordinate vertical axis shows the medicinal plants *Genus species* (used part, wild/cultivated)—UV (%); (1) winter tea: oregano, rose hips, leaves of strawberry and blackberries, (2) combined tea: oregano, elderberry, raspberry and blackberry leaf, (3) combined tea: oregano, lemon balm, Greek mountain tea, hawthorn, rosehip, raspberry and blackberry.

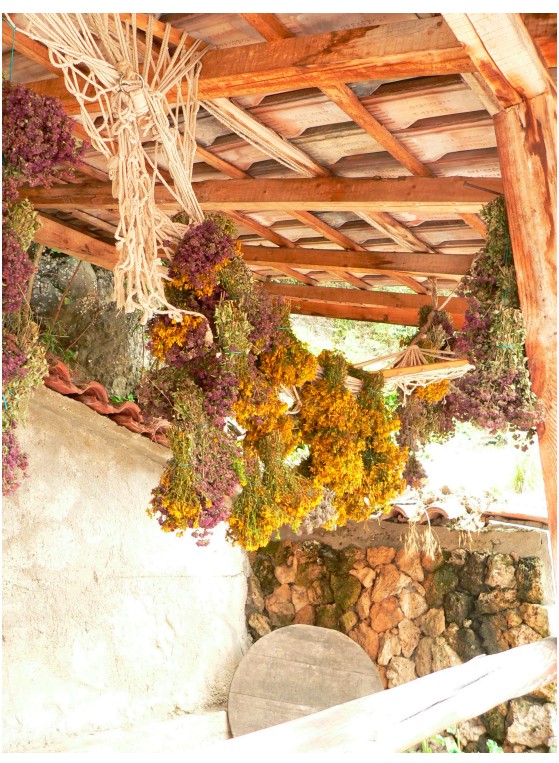

**Figure 7.** Collection of herbs in the Central Rhodopes—*O. vulgare* subsp. *vulgare* and *Hypericum perforatum*.

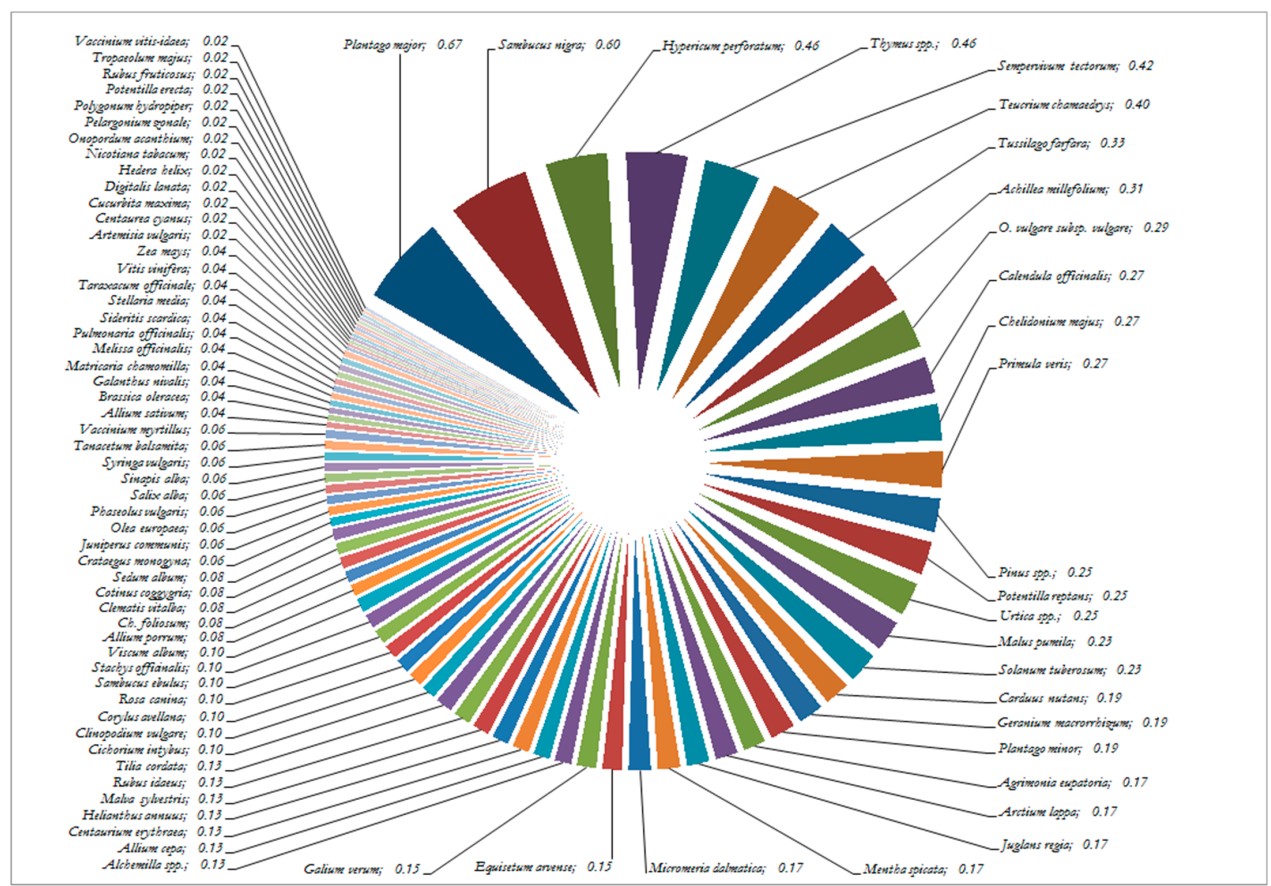

**Figure 8.** Plant species used for human health conditions in Central Rhodopes according to the value of the UV index.

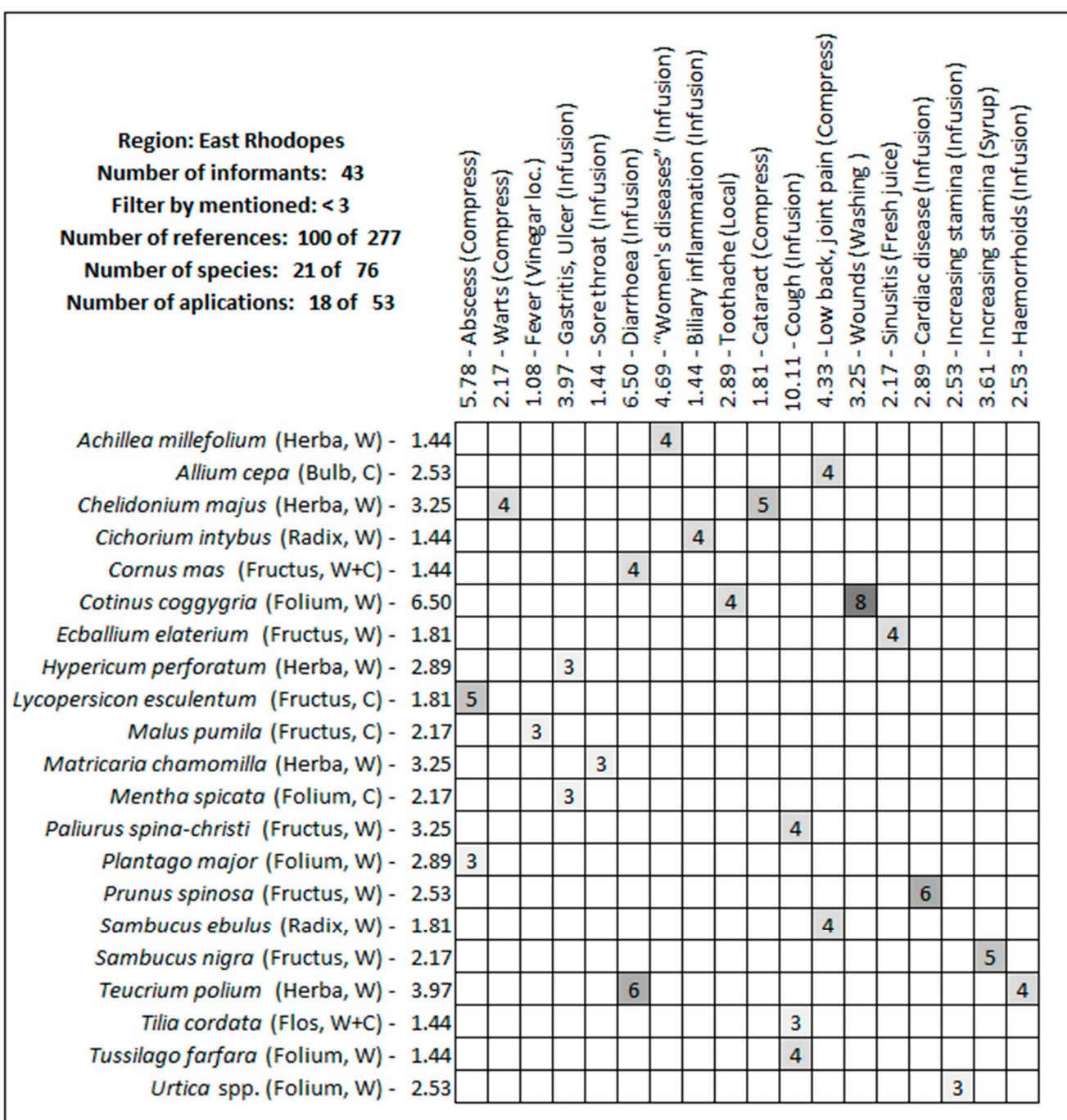

**Figure 9.** Nomogram of the medicinal plants for human use (at **ΣU** > 2) in the East Rhodopes, Legend: abscissa/horizontal axis of the nomograms depicts the conditions for which the informants apply the medicinal plants, as well as the way of processing the plant raw material; ordinate vertical axis shows the medicinal plants *Genus species* (used part, wild/cultivated)—UV (%).

Figure 10 presents a nomogram with a prophylactic application of medicinal plants in the East Rhodopes. Sixteen plant species are used to make tea, with the most reported being *Tilia cordata* (15), *Hypericum perforatum* (13) and *O. vulgare* subsp. *hirtum* (12).

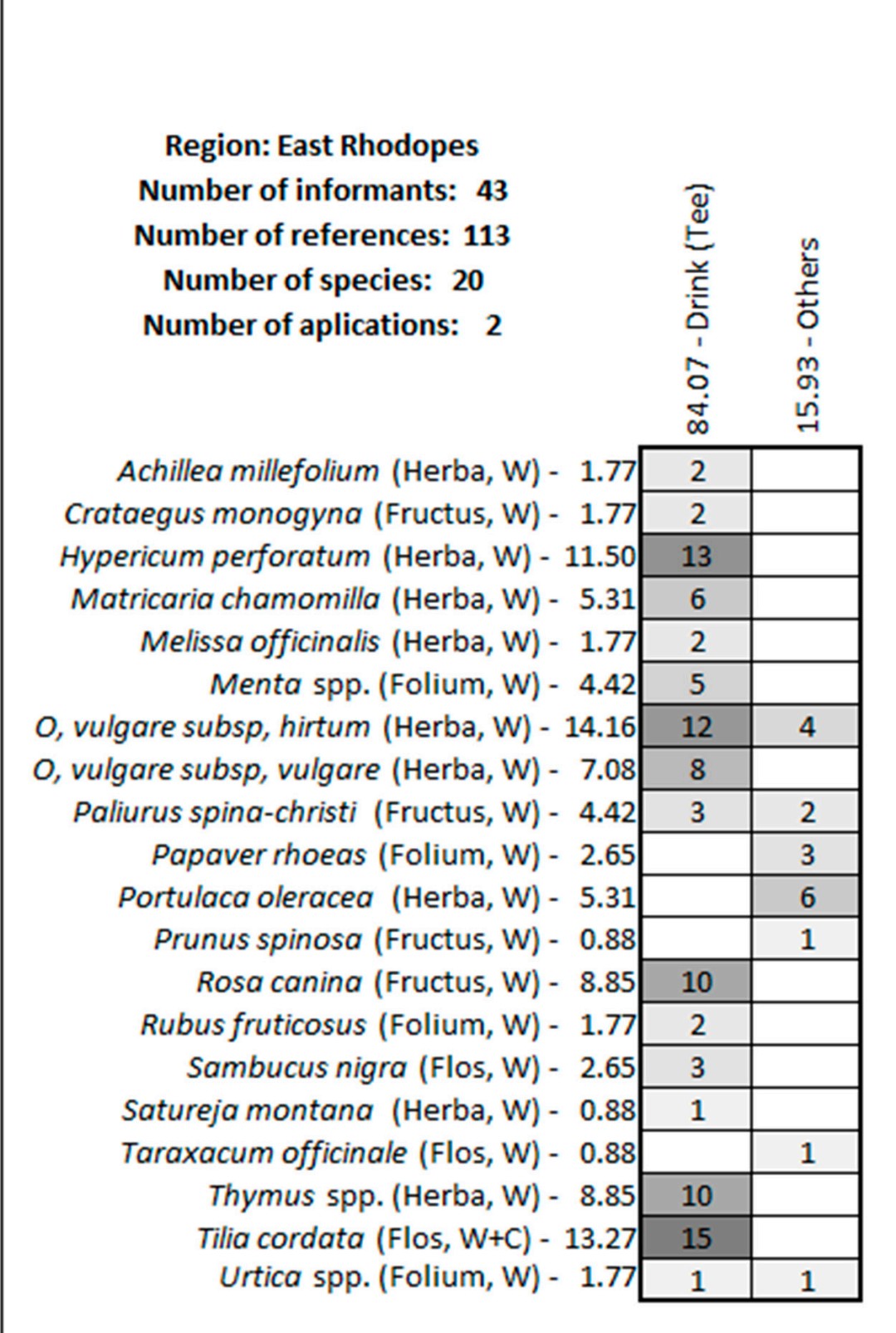

**Figure 10.** Nomogram of the medicinal plants with prophylactic application in the East Rhodopes Legend: abscissa/horizontal axis of the nomograms depicts the conditions for which the informants apply the medicinal plants, as well as the way of processing the plant raw material; ordinate vertical axis shows the medicinal plants *Genus species* (used part, wild/cultivated)—UV (%).

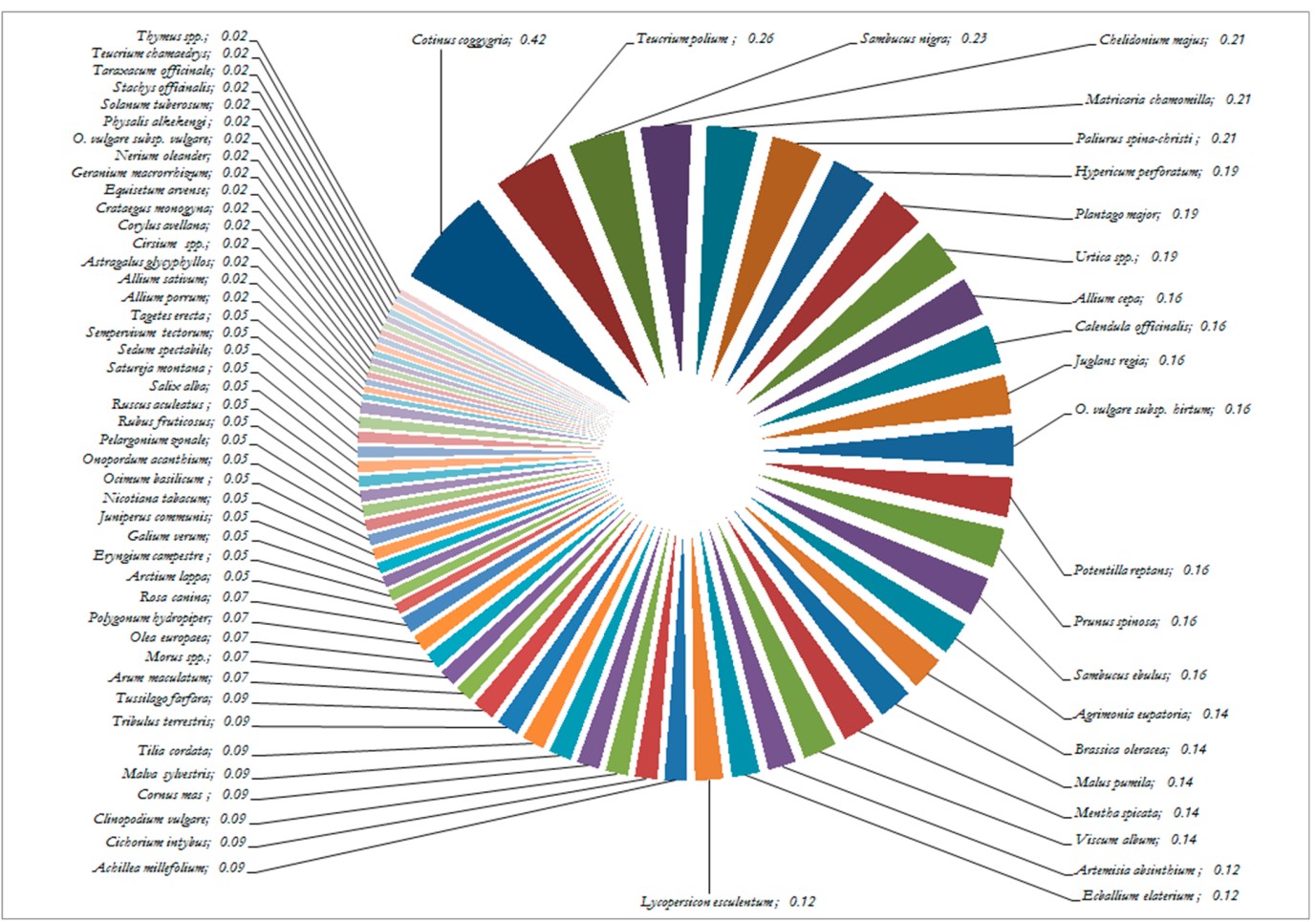

**Figure 11.** Plant species used for human health conditions in the East Rhodopes according to the value of the UV index.

## 4. Discussion

The highest degree of use (UV index) in the Central Rhodopes was found to be *Plantago major* (0.67), *Sambucus nigra* (0.60), *Hypericum perforatum* (0.46) and *Thymus* spp. and in the East Rhodopes, *Cotinus coggygria* (0.42), *Teucrium polium* (0.26), *Sambucus nigra* (0.23), *Chelidonium majus* (0.21), *Matricaria chamomilla* (0.21) and *Paliurus spina-christi* (0.21). Such high UV index was reported for *P. major, H. perforatum* and *Thymus* spp. in Suva Planina Mts., Serbia where *C. coggygria P. spina-christi* have not been reported [56].

Some particularities deserve attention. Surprising is the application of *Prunus spinosa* fruits against cardiac disease. This is based on what responders say. We cannot be sure about the extent to which their diagnosis would match an official medical diagnosis. In traditional Bulgarian medicine, it is very frequent for the herbs recommended for "sick heart" to relieve stomach disturbances in reality. This might be the case here. Another explanation might be the very high phenolate content of the fruits of *P. spinosa*. These phenolic compounds are known for their protective role against atherosclerosis. Thus, this knowledge might have somehow migrated from mainstream science to traditional lore. In Bulgarian traditional medicine, this plant substance is used as astringent [9,13,34]. In Greece, it is popular as a respiratory ailment [57] and in Turkey as a cure against renal disease and bronchitis [58,59].

Similarities between traditional medicine in the East Rhodopes and Turkey were expected but not convincingly confirmed [60–65]. For example, our finding of the application of the fruits of *Paliurus spina-christi* against cough is popular in Bulgaria [13,34]. It was recorded in Turkey to treat sore throat or bronchitis [60–64] but this plant has a broader

application there as a cure for kidney stones and diabetes [58,64,65] diarrhea and heart disease [60,62] or not reported at all [64].

Different plants are often used for one and the same disease in the Central and East Rhodopes. For example, the most often reported remedy for diarrhea is *Theucrium chamaedris* in the Central Rhodopes, while in the East Rhodopes it is *Theucrium polium*. It is the same with *O. vulgare* subsp. *Vulgare* and *O. vulgare* subsp. *Hirtum*. This can be explained with the specifics of the flora and vegetation of the two Rhodopes sub-regions. These settlements where the research took place (Figure 1) are surrounded by typical plant communities. The plant communities in the Central Rhodopes are of various types: Acidophilic spruce forests (Vaccinio-Piceetea); Beech forests of the type Luzulo-Fagetum dominated by *Fagus sylvatica, with Abies alba, Picea abies, Luzula luzuloides, Lerchenfeldia flexuosa, Calamagrostis arundinacea, Vaccinium myrtillus, Pteridium aquilinum*, etc.; Open oak forest dominated by *Quercus pubescens* and with participation of *Q. virgiliana, Q. frainetto, Q. cerris, Fraxinus ornus, Acer monspessulanum, Carpinus orientalis, Pistacia terebinthus*; Endemic oro-Mediterranean plant communities of spiny dwarf shrubs. Our field study covers Natura 2000 protected areas BG0002113 and BG0001031 in the Central Rhodopes. In the East Rhodopes, Mediterranean influence is detected. The largest area is occupied by mixed deciduous xerothermic forests of *Quercus cerris, Q. frainetto, Q. pubescens or Q. dalechampii* with Mediterranean and SubMediterranean elements, such as *Juniperus oxycedrus, Colutea arborescens, Carpinus orientalis* and others like *Paliurus spina-christi, Jasminum fruticans*, in combination with xerophytic Mediterranean formations. The percentage of open grassland with xerothermic and alluvial-meadow (around the river) grass formations is significant, as well as those of the shrub communities with Mediterranean elements and agricultural land.

Most of the plants have wide distribution and abundant populations. Our field study covered six Natura 2000 protected areas [42,43] and many plants in the area have conservation status [38]. The analysis of the obtained ethnobotanical data is valuable for the conservation and sustainable use of the medicinal plants. Among the endemic and/or rare plants with conservation status mentioned by the informants are *Lilium rhodopaeum, Orchis* sp. div., *Galanthus nivalis, G. elwesii, Sideritis scardica, Origanum vulgare subsp. Hirtum* and *Clinopodium dalmaticum* (Table 1).

*Lilium rhodopeum* Delip. is a critically endangered (IUCN) Balkan endemic [66], and protected by the Biological Diversity Act (2000, 2007) plant. The local people in the Central Rhodopes mentioned this plant (Table 1) only in relation to its conservation status, demonstrating their awareness but not about its medicinal application.

Our study revealed that *Leucojum aestivum,* which is partially protected by the Biodiversity Act [67] (Table 1), is used sporadically for ornamental purpose in the East Rhodopes (Madzarovo, Figure 1).

The "salep" orchids (at least 30 species from the genera *Aceras, Anacamptis, Barlia, Dactylorhiza, Himantoglossum, Ophrys, Orchis,* and *Serapias*) and several of them such as *Dactylorhiza kalopissii* E.Nelson and *Orchis spitzelii* Saut. ex W.D.J.Koch, *O. militaris* L. are extremely rare with single populations. All of the "salep" are listed in CITES and forbidden for international trade, 30 species in Bulgaria are under protection of the Biological Diversity Act (2000, 2007). Also, although 19 species are considered medicinal, according to the Medicinal Plant Act (2000) in Bulgaria their collecting is forbidden. The protection measures are not always completely efficient, but our ethnobotanical study revealed that nowadays "salep" is not popular amongst the local people in the Rhodopes and especially in the Central Rhodopes. However, monitoring is necessary to prevent damages, in case of future increased "salep" collection because the orchid population is vulnerable there [31].

This study reveals that even though both species of snowdrops, *Galanthus nivalis* and *G. elwesii* are considered endangered (IUCN) [68] and protected by the Biodiversity Act (2002) [67], the local people use the bulbs for medicinal purposes but only those of traditionally cultivated plants in their gardens.

*Sideritis scardica*, which is an endangered (IUCN) Balkan endemic plant species [69] and forbidden for collection from wild populations for trading (Medicinal Plant Act 2002) [1]

(Table 1), has become rather popular during the last decades in Bulgaria causing the vulnerability of its wild populations [70]. Our research shows that there are many cultivated plots in the Central Rhodopes (e.g., Trigrad, Chepelare, Mugla, Momchilovtzi) both on a small scale in personal kitchen gardens and on a bigger scale for trading purposes. This can be regarded as a good practice for conservation of this species.

Special attention and further actions for efficient cultivation need the Balkan endemics *Clinopodium dalmaticum* and *Haberlea rhodopensis*. *C. dalmaticum* appears to be a locally popular and actively collected medicinal plant and this is evaluated as a hazard for its wild populations (Table 1). Although the plant is not a "Red Data Book species", it is one of the characteristic taxa of the Red Data Book habitats, categorized as vulnerable, namely "Ultra-basic rocks with pioneer herbaceous vegetation" [71]. *H. rhodopensis* is not a locally popular medicinal plant and therefore not collected by the informants. Despite the fact that it is protected by the Biodiversity Act (2002) [67], an increasing interest for cosmetic and pharmaceutical industry is focused on it with several commercial products.

*Origanum vulgare* subsp. *hirtum* is restricted to the western Mediterranean as well as to a few localities in the Southernmost parts of Bulgaria (Table 1). Therefore, the plant is forbidden for collection for trade purposes in Bulgaria according to the Medicinal Plants Act (2002) [1]. Our ethnobotanical research reveals that although there is a strong tradition of collecting *O. vulgare* subsp. *hirtum*, currently the pressure on the wild populations is reduced due the tendency of depopulation those villages in the East Rhodopes where the plant grows [32].

Ethnobotanical studies of the traditional knowledge and use of medicinal plants is a way to discover phylogeny-guided drugs in the early screening stage, which may lead to a higher discovery efficiency of new drugs with meaningful biological activities and with big economical potential [72–74].

Additionally, our experience in the filed study confirms the necessity of a document which defines guidelines for best practice on how to conduct and report ethnopharmacological studies [75,76].

## 5. Conclusions

The rural population in the Rhodopes knows and uses a high number of plants for medicinal purposes and the tradition is still alive despite the tendency towards globalization, particularly in the remote villages that rely on plants to treat various health problems. Many of the plants are used in both sub-regions, the Central and East Rhodopes, but also some peculiarities are detected for each of them. The Central Rhodopes has the highest degree of use with the *Plantago major* (0.67), *Sambucus nigra* (0.60), *Hypericum perforatum* (0.46), *Thymus* spp. (0.46), *Sempervivum tectorum* (0.42), *Teucrium chamaedrys* (0.40), *Tussilago farfara* (0.33), *Achillea millefolium* (0.31) and *Origanum vulgare* subsp. *Vulgare* (0.29). Most often mentioned in the East Rhodopes were *Cotinus coggygria* (0.42), *Teucrium polium* (0.26), *Sambucus nigra* (0.23), *Chelidonium majus* (0.21), *Matricaria chamomilla* (0.21), *Paliurus spina-christi* (0.21), *Plantago major* (0.19) and *Urtica* spp. (0.19). These differences are explained mostly by the differences of the phyto-climate in the two sub-regions. The vegetation in the Central Rhodopes has boreal and Central European influence, while in the East Rhodopes, the Mediterranean influence is significant. Interestingly different plants are often used for one and the same disease in the Central and East Rhodopes. Similarities between traditional medicine in the East Rhodopes and Turkey were expected but not convincingly confirmed. The rural population in the Rhodopes use the medicinal plants sustainably, and they collect widespread plants. Not much anthropogenic pressure was detected on the rare plants such as *Lilium rhodopeum* and "salep" orchids. Furthermore, the locals are open to the cultivation of *Sideritis scardica*—a species, which became rare after overexploitation in the last decades. The hazard is for *Clinopodium dalmaticum*, which has recently been actively collected and requires an introduction to cultivation. The hazard for *Haberlea rhodopensis* is not from the locals and their traditional way of medicinal plants' application, but by the industry, and this case needs caution monitoring and control by the authorities.

**Supplementary Materials:** The following supporting information can be downloaded at: https://www.mdpi.com/article/10.3390/d14080686/s1, Figure S1: The full nomogram of data for the application of medicinal plants classified by the degree of their usability, namely for treating human health conditions; Figure S2: The full nomogram of data for the usage of medicinal plants classified by the degree of their application, namely for treating human health conditions.

**Author Contributions:** Conceptualization, I.M. and E.K.; methodology, I.M., C.R. and E.K.; validation, L.R., I.A., Z.N. and E.K.; formal analysis, I.M., C.R. and E.K.; investigation, I.M.; writing—original draft preparation, I.M.; writing, review, and editing, E.K., Z.N., I.A. and L.R.; visualization, I.M., C.R., E.K.; supervision, E.K.; project administration, I.A. All authors have read and agreed to the published version of the manuscript.

**Funding:** Ina Aneva expresses her thanks for the support provided by the Bulgarian Ministry of Education and Science under the National Research Programme "Healthy Foods for a Strong Bio-Economy and Quality of Life" approved by DCM № 577/17.08.2018.

**Institutional Review Board Statement:** The study was conducted according to the guidelines of the Declaration of Helsinki, and approved by the Animal Care Ethic Committee of the Bulgarian Agency for Food Safety (BAFS) (protocol code 187 from 13 November 2017).

**Data Availability Statement:** Experimental data are available from the corresponding author upon reasonable written request.

**Conflicts of Interest:** The authors declare no conflict of interest.

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
