# Peer review of "Ethnobotanical and Ethnopharmacological Study in the Bulgarian Rhodopes Mountains—Part _I"

_diversity, doi:10.3390/d14080686_

Round 1
Reviewer 1 Report
This article present Ethnobotanical and ethnopharmacological study in the Bulgarian Mountain Rhodopes. This study will help to facilitate to understand medicinal uses and distribution of medicinal plants in a specific region. Before recommending this article for publication, there are some shortcomings for that should be resolve.
General comments
Overall, the study is well designed and presented in a good way, but mostly the literature is not cited. Grammatical and typos must be revised
Abstract
Revise the sentence “and their use has” their uses have been.
Specify the methods in the abstract.
Remove the note and table cited in the abstract
Add one or two lines conclusion and future perspective of the study.
The methodology should be brief and clear.
Also add quantitative results in this section.
Introduction
The introduction part is well written but still some details are required. The authors should provide details of the distribution of the studied species in the world as well as in Bulgaria.
Threats to conservation of medicinal plants.
Cite first sentence of the introduction.
Discuss Specific health related issues in Rhodopes for which ethnobotanical remedies are considered.
Discuss economic importance and advance approaches for discovering ethnobotanical plants by citing the following literature.
http://dx.doi.org/10.30848/PJB2022-3(19), https://doi.org/10.1016/j.jep.2021.114515,
Materials and methods
Methodology is well written. Replace “ware” with was Line 89.
Write section number “ Data visualization by nomograms”
Results
Results are well presented but revise grammatical and typo mistakes.
Discussion
Compare the obtained with more current study.
Discussion must be more elaborative. It would be better to discuss every result and compare with current studies
Conclusion
Conclusion is well justified.
Author Response
Reviewer 1
Comments and Suggestions for Authors and Point by point comments by the Authors
This article present Ethnobotanical and ethnopharmacological study in the Bulgarian Mountain Rhodopes. This study will help to facilitate to understand medicinal uses and distribution of medicinal plants in a specific region. Before recommending this article for publication, there are some shortcomings for that should be resolve.
Thank you to Reviewer 1 for the detailed and useful comments
General comments
Overall, the study is well designed and presented in a good way, but mostly the literature is not cited. Grammatical and typos must be revised
Abstract
Revise the sentence “and their use has” their uses have been.
Point 1 corrected
Specify the methods in the abstract.
Point 2 corrected
Remove the note and table cited in the abstract
Point 3 corrected
Add one or two lines conclusion and future perspective of the study.
The methodology should be brief and clear.
Also add quantitative results in this section.
Point 4 corrected
Introduction
The introduction part is well written but still some details are required. The authors should provide details of the distribution of the studied species in the world as well as in Bulgaria.
Threats to conservation of medicinal plants.
Cite first sentence of the introduction.
Discuss Specific health related issues in Rhodopes for which ethnobotanical remedies are considered.
Discuss economic importance and advance approaches for discovering ethnobotanical plants by citing the following literature.
http://dx.doi.org/10.30848/PJB2022-3(19) , https://doi.org/10.1016/j.jep.2021.114515,
Point 5 All comments above are reflected in the MS and text is corrected. We citated the suggested literature, and we found its place in the discussion part, where we consider it most relevant to the text.
Materials and methods
Methodology is well written. Replace “ware” with was Line 89.
Write section number “Data visualization by nomograms”
Point 6 corrected -
Results
Results are well presented but revise grammatical and typo mistakes.
Point 7 corrected -
Discussion
Compare the obtained with more current study.
Discussion must be more elaborative. It would be better to discuss every result and compare with current studies
Point 7 corrected
Conclusion
Conclusion is well justified.

Reviewer 2 Report
This is a descriptive quantitative study about well-known plant drugs (repeatedly reported in the literature since antiquity) and general plant use patterns of species used in traditional medicine in a rural region of Bulgaria. It is clear, that comparing quantitative data and values obtained with unscientific indices will always lead to the insight that use(s) (values) are either lower or higher when compared to other locations. However, the data provided herein is not even compared to other studies and thus the novelty does not become clear. The language needs substantial improvement.
The introduction is too general and unfocused. Instead, it should focus on regional herbal medicine, cultural and ethnopharmacological background, available health care options in the area of research and epidemiology, diagnostics and provide a theoretical background for your research question.
In ethnomedicinal field research asking for specific remedies when conducting a general survey is not considered adequate. You should simply ask: which remedies/plants/animals/minerals and other resources do you use for preventing and curing diseases and illnesses?
Herbarium vouchers need to be intercalated in a recognized herbarium so that the hard data can be verified.
Classifying diseases according to the ICD-10 depends on a medical diagnosis with the help of diagnostic tools and cannot be applied to post-hoc classifications. Not even medical doctors would be able to make such a classification. Normally the emic/local disease classification is applied otherwise the ICPC-2 (also by the WHO) can be used (DOI: 10.1016/j.jep.2015.08.051)
The UV is a % value of doubtful relevance for ethnopharmacology or ethnomedicine as it includes all uses and is statistically wrong because the variance of the data is not included. The number of use-reports gathered for the specific indications is enough for understanding the versatility of plant use. The number of use-reports gathered is rather low.
Nomograms can make sense when they indicate significant overuse of a specific drug for a certain use-category but when this opportunity (that can be done in “R”) is not used the simple listing of plant drugs, plant part used, mode of application and specific therapeutic use is more adequate and straightforward.
In the conclusions no citations should appear (otherwise it belongs to the discussion section).
The particularities should be worked out and explained.
For more information and further references please check: Heinrich M, Lardos A, Leonti M, Weckerle C, Willcox M; with the ConSEFS advisory group; Based on a consultative process of researchers active in ethnopharmacology and with particular input by the ConSEFS Advisory group:, Applequist W, Ladio A, Lin Long C, Mukherjee P, Stafford G. Best practice in research: Consensus Statement on Ethnopharmacological Field Studies - ConSEFS. J Ethnopharmacol. 2018
Author Response
Reviewer 2
Comments and Suggestions for Authors and Point by point comments by the Authors
This is a descriptive quantitative study about well-known plant drugs (repeatedly reported in the literature since antiquity) and general plant use patterns of species used in traditional medicine in a rural region of Bulgaria. It is clear, that comparing quantitative data and values obtained with unscientific indices will always lead to the insight that use(s) (values) are either lower or higher when compared to other locations. However, the data provided herein is not even compared to other studies and thus the novelty does not become clear. The language needs substantial improvement.
First of all, thank you to Reviewer 2 for the detailed and useful comments
Point 1 corrected - compared to more studies (comparison was done with similar studies from Turkey)
The introduction is too general and unfocused. Instead, it should focus on regional herbal medicine, cultural and ethnopharmacological background, available health care options in the area of research and epidemiology, diagnostics and provide a theoretical background for your research question.
Point 2 Concerning the available health care it was originally stated in the text “our preliminary observation in a number of settlements revealed that even nowadays the local population particularly in the winter months relies solely on the medicinal plants collected during the summer”.
Local ethnobotanical studies on medicinal plants in the region are not available except the cited ones. Epidemiology data for the region are added even though few research is available
http://www.medicalbiophysics.bg/en/3-2018-BG-JournalPH.html
https://europepmc.org/article/med/3414101
In ethnomedicinal field research asking for specific remedies when conducting a general survey is not considered adequate. You should simply ask: which remedies/plants/animals/minerals and other resources do you use for preventing and curing diseases and illnesses?
Point 3 This is exactly the way that we asked the questions – and it was a semistructured interview we only used broad organ- and therapy-based use-categories following roughly the WHO, 2018. World Health Organization. International classification of diseases 2018. Geneva, Switzerland
Herbarium vouchers need to be intercalated in a recognized herbarium so that the hard data can be verified.
Point 4 We keep herbarium material obtained as a result from the field study in personal collection as well as photographs of the plants reported by the informants.
Classifying diseases according to the ICD-10 depends on a medical diagnosis with the help of diagnostic tools and cannot be applied to post-hoc classifications. Not even medical doctors would be able to make such a classification. Normally the emic/local disease classification is applied otherwise the ICPC-2 (also by the WHO) can be used (DOI: 10.1016/j.jep.2015.08.051)
Point 5 We agree with your remark and our filed study was done as a semi-structured interview with only approximately pointing the broad organ- and therapy-based use-categories following roughly the WHO, 2018. We corrected the text according to your remark
The UV is a % value of doubtful relevance for ethnopharmacology or ethnomedicine as it includes all uses and is statistically wrong because the variance of the data is not included. The number of use-reports gathered for the specific indications is enough for understanding the versatility of plant use. The number of use-reports gathered is rather low.
Point 6 Since this index is widely used and it gives possibilities for comparison of the results, we chose to use it too.
Nomograms can make sense when they indicate significant overuse of a specific drug for a certain use-category but when this opportunity (that can be done in “R”) is not used the simple listing of plant drugs, plant part used, mode of application and specific therapeutic use is more adequate and straightforward.
Point 6 We find nomograms useful for visualization of the data
In the conclusions, no citations should appear (otherwise it belongs to the discussion section).
Point 7 Corrected in accordance with this remark
The particularities should be worked out and explained.
Point 8 Corrected in accordance with this remark
For more information and further references please check: Heinrich M, Lardos A, Leonti M, Weckerle C, Willcox M; with the ConSEFS advisory group; Based on a consultative process of researchers active in ethnopharmacology and with particular input by the ConSEFS Advisory group:, Applequist W, Ladio A, Lin Long C, Mukherjee P, Stafford G. Best practice in research: Consensus Statement on Ethnopharmacological Field Studies - ConSEFS. J Ethnopharmacol. 2018
Point 9 This is an extremely useful publication. Unfortunately, when we actively collected the main body of the data in the field this valuable work was not published yet.
Publications cited
- Heinrich, M., Lardos, A., Leonti, M., Weckerle, C., Willcox, M., Applequist, W., ... & Stafford, G. (2018). Best practice in research: consensus statement on ethnopharmacological field studies–ConSEFS.. J Ethnopharmacol. 2018, 211: 329-339 doi: 10.1016/j.jep.2017.08.015
- Weckerle CS, de Boer HJ, Puri RK, van Andel T, Bussmann RW, Leonti M. Recommended standards for conducting and reporting ethnopharmacological field studies. J Ethnopharmacol. 2018, 210:125-132. doi: 10.1016/j.jep.2017.08.018

Reviewer 3 Report
Dear editor
In my opinion paper is well written. However some revisions seems necessary
A comprehensive literature review on similar studies seems to be necessary in introduction
A perfect view on studied area is necessary including geology, geomorphology, climatology, culture, population and other social aspects
Please provide some images on general view of habitats as well the structure the area
The images of important plant taxa is necessary
Scientific names should be cited by author at the first time
Discussion should be improved including:
Comparison between mentioned studies to other studies in diffident regions.
Main achievements the study?
What are the conservation issue of study?
Please clearly discuss about them
Some sections should be transferred to discussion
Conclusion is summary of main achievements, please reduce it
Best Regards

Author Response
Reviewer 3
Comments and Suggestions for Authors
Dear editor
In my opinion paper is well written. However some revisions seems necessary
A comprehensive literature review on similar studies seems to be necessary in introduction
Point 1 Corrected in accordance to this remark
A perfect view on studied area is necessary including geology, geomorphology, climatology, culture, population and other social aspects
Please provide some images on general view of habitats as well the structure the area
The images of important plant taxa is necessary
Point 2 We have too many illustrations therefore additional images will be too many. Images of all plant taxa mentioned in this research are available on line.
Scientific names should be cited by author at the first time
Point 3. Scientific names should be cited by author in Table 1 which for some technical reason was not visible in the body text imported in the template but which was also attached separately. On the submission
Discussion should be improved including:
Comparison between mentioned studies to other studies in diffident regions.
Main achievements the study?
What are the conservation issue of study?
Please clearly discuss about them
Some sections should be transferred to discussion
Conclusion is summary of main achievements, please reduce it
Point 4 All corrected in accordance with this remark
